# Biosorption of Pb(II) Using Natural and Treated *Ardisia compressa* K. Leaves: Simulation Framework Extended through the Application of Artificial Neural Network and Genetic Algorithm

**DOI:** 10.3390/molecules28176387

**Published:** 2023-08-31

**Authors:** Alma Y. Vázquez-Sánchez, Eder C. Lima, Mohamed Abatal, Rasikh Tariq, Arlette A. Santiago, Ismeli Alfonso, Claudia Aguilar, América R. Vazquez-Olmos

**Affiliations:** 1Área Agroindustrial Alimentaria, Universidad Tecnológica de Xicotepec de Juárez, Av. Universidad Tecnológica No. 1000. Col. Tierra Negra Xicotepec de Juárez, Puebla 73080, Mexico; alma.vazquez@utxicotepec.edu.mx; 2Institute of Chemistry, Federal University of Rio Grande do Sul (UFRGS), Av. Bento Goncalves 9500, P.O. Box 15003, Porto Alegre 91501-970, RS, Brazil; eder.lima@ufrgs.br; 3Facultad de Ingeniería, Universidad Autónoma del Carmen, Campeche 24115, Mexico; 4Institute for the Future of Education, Tecnologico de Monterrey, Monterrey 64849, Mexico; rasikhtariq@gmail.com; 5Escuela Nacional de Estudios Superiores, Unidad Morelia, Universidad Nacional Autónoma de México, Antigua Carretera a Pátzcuaro No. 8701, Col. Ex. Hacienda de San José de la Huerta, Morelia 58190, Mexico; arlette_santiago@enesmorelia.unam.mx; 6Instituto de Investigaciones en Materiales, Unidad Morelia, Universidad Nacional Autónoma de México, Antigua Carretera a Pátzcuaro No. 8701, Col. Ex. Hacienda de San José de la Huerta, Morelia 58190, Mexico; ialfonso@iim.unam.mx; 7Facultad de Química, Universidad Autónoma del Carmen, Calle 56 No. 4 Av. Concordia, Ciudad del Carmen, Campeche 24180, Mexico; caguilar@pampano.unacar.mx; 8Instituto de Ciencias aplicadas y Tecnología, UNAM, Circuito Exterior, S/N, Ciudad Universitaria, A.P. 70-186, Delegación Coyoacán, Ciudad de México 04510, Mexico; america.vazquez@icat.unam.mx

**Keywords:** *Ardisia compressa* K., biosorption, heavy metals, artificial neural network, educational Innovation

## Abstract

This study explored the effects of solution pH, biosorbent dose, contact time, and temperature on the Pb(II) biosorption process of natural and chemically treated leaves of *A. compressa* K. (Raw-AC and AC-OH, respectively). The results show that the surface characteristics of Raw-AC changed following alkali treatment. FT-IR analysis showed the presence of various functional groups on the surface of the biosorbent, which were binding sites for the Pb(II) biosorption. The nonlinear pseudo-second-order kinetic model was found to be the best fitted to the experimental kinetic data. Adsorption equilibrium data at pH = 2–6, biosorbents dose from 5 to 20 mg/L, and temperature from 300.15 to 333.15 K were adjusted to the Langmuir, Freundlich, and Dubinin–Radushkevich (D-R) isotherm models. The results show that the adsorption capacity was enhanced with the increase in the solution pH and diminished with the increase in the temperature and biosorbent dose. It was also found that AC-OH is more effective than Raw-AC in removing Pb(II) from aqueous solutions. This was also confirmed using artificial neural networks and genetic algorithms, where it was demonstrated that the improvement was around 57.7%. The nonlinear Langmuir isotherm model was the best fitted, and the maximum adsorption capacities of Raw-AC and AC-OH were 96 mg/g and 170 mg/g, respectively. The removal efficiency of Pb(II) was maintained approximately after three adsorption and desorption cycles using 0.5 M HCl as an eluent. This research delved into the impact of solution pH, biosorbent characteristics, and operational parameters on Pb(II) biosorption, offering valuable insights for engineering education by illustrating the practical application of fundamental chemical and kinetic principles to enhance the design and optimization of sustainable water treatment systems.

## 1. Introduction

Over recent decades, heavy metals have been monitored in sediments, soils, and water systems in different regions of the world in order to study their risks to wildlife and human health [1,2,3]. In these studies, the reported concentrations of various toxic metals, such as chromium (Cr), mercury (Hg), lead (Pb), nickel (Ni), and cadmium (Cd), exceed the permissible limits according to the United States Environmental Protection Agency (US-EPA) and World Health Organization (WHO) [1]. In the previous investigations, urban sewage and effluents discharged from different manufacturers like pigment, battery, and pesticides or mining activities are considered the principal sources of these hazardous metals in the monitored sites [1]. The presence of toxic metals, even at low concentrations in soils and water, can cause severe animal and human health risks, even cancer and other diseases, such as respiratory irritation, bronchial spasms, and coughing [4,5,6]. Therefore, intense investigations have developed various industrial and urban wastewater treatment techniques, including coagulation, chemical precipitation, reverse osmosis, ion exchange, flocculation, and membrane filtration, that are used before the waste products are discharged into the aquatic system [7]. Among these methods, adsorption has become the most used technique for removing organic or inorganic pollutants due to its simplicity, high efficiency, and low cost [8]. Numerous adsorbents have been developed for Pb(II) ion removal, such as zeolites [9], kaolinite clay [10], carbon-based materials [11], agricultural waste [12], and polymeric materials [13]. The cost and availability of the adsorbent materials are crucial to the efficiency of the adsorption technique. In this regard, many agriculture and industrial waste materials have been widely used to remove heavy metals from wastewater due to their low cost, availability, and the possibility for generation over various adsorption/desorption cycles [14,15,16,17]. A number of agro-wastes were studied for the adsorption of Pb (II) from aqueous solution, such as *Azadirachta indica* leaves [18], *Tamarindus indica* seeds [19], and rape straw [20], as a result of their chemical components namely, steroids, saponins, flavonoids, alkaloids, tannins, and amino acid [21]. Recent investigations reported that chemical modification increased the number of binding ligand groups on the biosorbent’s surface. Therefore, the adsorption capacity of biosorbents is expected to be enhanced because the heavy metal ion uptake is favored by increasing active sites on the biosorbent’s surface [22].

*A. compressa K*. belongs to the Myrsinaceae family, which is found in tropical and subtropical regions and has been used as food and folk medicine in different parts of the world [23]. In Mexico, *A. compressa K*. is found in different states, such as Aguascalientes, Chiapas, Chihuahua, Guerrero, Hidalgo, Jalisco, Michoacán, Nayarit, Oaxaca, Puebla, Tlaxcala, San Luis Potosí, Sinaloa, Tamaulipas, and Veracruz [24]. Notably, in the North of Puebla, it is known as *Acachul*, and recent investigations reported by Vázquez-Sánchez A.Y. et al. [24] and by Joaquín-Cruz, Elvia et al. [23] showed that *A. compressa K*. fruits contain various phytochemicals like anthocyanins, polyphenols, flavonoids, diterpenes, gallotannins, and chlorogenic acids. In contrast, Chandra and Mejia [25] identified the presence of phenolic constituents, such as catechin, epicatechin gallate, gallic acid, kaempferol, and ardisin, in the aqueous extracts of the leaves of *A. compressa K*. Therefore, *A. compressa K*. biomass can be considered as an alternative potential biosorbent for heavy metal removal from water due to its interesting properties and its availability.

Another novel aspect of this research is to resolve such a multivariate problem to find the optimal combination of all the tuning variables for the maximization of the removal fraction. This is attained through the application of artificial neural networks for the regression model and the genetic algorithm for optimization purposes. 

To the best of our knowledge, *A. compressa* K. biomass has not been used as a biosorbent. Thus, this work contributes to studying the biosorption capacity of natural and treated-with-NaOH *A. compressa* K. leaves regarding removing Pb(II) from aqueous solutions varying solution pH, contact time, adsorbent dose, and temperature. In addition, the adsorption–desorption cycle experiments studied the reuse of raw and treated biosorbents.

## 2. Results and Discussion

### 2.1. Physicochemical Characterization

Table 1 presents the physicochemical parameters of *A. compressa* K. leaves in a sense to determine their nutritional composition. The results are in agreement with various reports of the proximate chemical composition of the other biosorbents, such as *Leucaena leucocephala* leaves [26,27] and *Nostoc commune* [28]. For example, if we compare our results with those reported for *Leucaena leucocephala* leaves, the ash content was higher than the 5.35% reported by Thamaga et al. (2021) [26] but lower than the 6.62% reported by G.W. Garcia et al. (2013) [27]; likewise, the protein contents were lower than 19.53% and 22.03% [26,27]. These variations could be attributed to differences in the ages of trees, agro-climatic conditions, and maturity stages [29]. Also, based on the results and in comparison, *N. commune* samples had a high moisture content (96.52 ± 0.11%) and high amounts of non-nitrogen-containing components (69.56 ± 0.09%) compared with *A. compressa* K. leaves. Furthermore, the values of the crude fat (0.24 ± 0.02%), crude fiber (3.42 ± 0.06%), and ash (5.25 ± 0.08%) were lower than *A. compressa* K. leaves [30]. 

### 2.2. Characterization

#### 2.2.1. Zero-Point Charge

Figure 1 shows the initial pH (pH_initial_) variation vs. the final pH (pH_final_) of Raw-AC and AC-OH biosorbents. As can be observed from Figure 1, the intersection of the diagonal line and the pH_initial_ vs. pH_final_ curves indicates that the pH_PZC_ values of Raw-AC and AC-OH were 4.8 and 6.4, respectively. The increase in the pH_PZC_ after the alkali treatment of Raw-AC confirmed the modification in its surface charge, which might improve the biosorption of metal ions since when pH > pH_PZC_, the surface becomes negative, whereas it is positive when pH < pH_PZC_. Similar results were obtained and are presented in Table 2, where the pH_PZC_ values of some raw biosorbents increased with alkaline treatment.

#### 2.2.2. Scanning Electron Microscopy

The chemical composition before and after contact with the Pb(II) solution of the RAW-AC and AC-OH samples was estimated using energy-dispersive X-ray (EDX) spectroscopy. Both samples show mainly the presence of carbon (C), oxygen (O), calcium (Ca), and aluminum (Al). The EDX analysis results show that the amount of Pb was 4.4 Wt% for the RAW-AC sample, whereas for the AC-OH sample, the Pb adsorbed was 7.5% Wt%. These results and their corresponding SEM images of RAW-AC and AC-OH before and after contact with Pb (II) are presented in Figure 2 and Figure 3, respectively. For both samples, the presence of Pb (II) is clear in the zoomed-in image, as represented by red dots.

#### 2.2.3. FTIR Analysis

Fourier transform infrared (FTIR) analysis of the biosorbent samples was performed to confirm the presence of functional groups. Figure 4 shows the bands corresponding to the identified functional groups in the biosorbents. The broad band, observed from 3310 to 3280 cm^−1^ for both the before-biosorption and after-biosorption samples, indicates the presence of -OH and -N-H groups [35]. The bands at 2920, 2900, and 2850 cm^−1^ represent the existence of methyl and methylene groups. The bands at 1730 and 1610 cm^−1^ represent the characteristics of the carbonyl group (C=O) stretching from aldehydes and ketones, which could be conjugated or non-conjugated to aromatic rings [36]. The bands at 1450 cm^−1^ show CH_2_ and CH_3_ groups, while the 1160 and 1220 cm^−1^ bands illustrate C-N [37]. The band at 1610 cm^−1^ was shifted to 1590 cm^−1^ after the Pb(II) uptake, suggesting the interaction of the carbonyl group with Pb(II) during the biosorption process. The *A. compressa* K. has a band at 1030 cm^−1^ and was assigned to the stretch of the C-O group of ether or ester. After the Pb(II) uptake, this band was shifted to 1020 cm^−1^, indicating the interaction of biomass with Pb(II) [38]. As expected, the surface of the biosorbents contains different functional groups, such as carboxyl, hydroxyl, and amino groups, which could be a potential adsorption site for biosorption.

### 2.3. Biosorption Results

#### 2.3.1. Effect of Solution pH

The effect of the solution pH on the Pb(II) sorption capacity of Raw-AC and AC-OH was studied using the isotherm experiments. Figure 5 shows the isotherm adsorption of Pb(II) for solution pHs at 2, 4, and 6. As observed, the Pb(II) sorption capacity of Raw-AC and AC-OH increased with the increase in adsorbate solution pH. This dependence can be associated with the metal species present at different pH solutions and the biosorbent’s surface charge [39].

The lower removal of Pb(II) at acidic pH (pH = 2 and 4) can be attributed to two reasons: (a) the competition between H^+^ present in the solution and Pb^2+^ ions that are the predominant species in this pH region, and (b) the electrostatic repulsion between the Pb^2+^ and the positively charged surface of Raw-AC and AC-OH biosorbents because the solution pH was lower than pH_PZC_(Raw-AC) = 4.8 and pH_PZC_(AC-OH) = 6.4. At pH = 6, the capacity of adsorption of biosorbents was superior when compared with pH = 2 and 4; this can be explained by the decrease in competition between protons and cations of Pb on the surface of the biosorbent due to the lower number of protons in the solutions. Furthermore, at pH = 6, the surfaces of biosorbents became negative, and the biosorption of Pb ions was favored over the ionic states of the carboxyl, hydroxyl, and amino groups [40].

Table 3 gives the parameter values of the Langmuir, Freundlich, and Dubinin–Radushkevich isotherm models using their non-linear forms (Equations (4), (5), and (6), respectively). Comparing the correlation coefficients (R^2^) obtained from the nonlinearized Langmuir, Freundlich, and Dubinin–Radushkevich isotherm models, the Langmuir model was the best fitted to the experimental equilibrium data.

For Raw-AC, at pH = 2, R^2^(Langmuir) = 0.981 > R^2^(Freundlich) = 0.960 > R^2^(Dubinin–Radushkevich) = 0.897; at pH = 4, R^2^(Langmuir) = 0.992 > R^2^(Freundlich) = 0.970 > R^2^(Dubinin–Radushkevich) = 0.932; and at pH = 6, R^2^(Langmuir) = 0.997 > R^2^(Freundlich) = 0.972 > R^2^(Dubinin–Radushkevich) = 0.910.

For AC-OH, at pH = 2, R^2^(Langmuir) = 0.971 > R^2^(Freundlich) = 0.924 > R^2^(Dubinin–Radushkevich) = 0.917; at pH = 4, R^2^(Langmuir) = 0.984 > R^2^(Freundlich) = 0.913 > R^2^(Dubinin–Radushkevich) = 0.867; and at pH = 6, R^2^(Langmuir) = 0.990 > R^2^(Freundlich) = 0.976, > R^2^(Dubinin–Radushkevich) = 0.913.

At pH = 2, 4, and 6, the maximum sorption capacities of Raw-AC determined from the Langmuir model (Q_m_) were 26.4, 48.1, and 96.4 mg/g, whereas, for AC-OH, the Q_m_ values were 43.7, 49.6, and 170.9 mg/g, respectively. As shown in Table 3, the values of K_L_ for both biosorbents decreased with the increase in solution pH (from 2.001.10^−2^ to 0.781.10^−2^ L/mg for Raw-AC and from 59.431.10–2 to 9.076 10^−2^ L/mg for AC-OH), indicating that the adsorption of Pb(II) was not favored in an acidic solution. Similar results were reported using other biosorbents [39,40].

The values of the average relative error (ARE), the sum of square error (SSE), normalized standard deviation Δq (%), chi-squared test (χ^2^), the sum of absolute error (EABS), and the root mean square error (RMSE) are presented in Table 4.

Comparing the values calculated for both isotherm models, it can be observed that the lower values of all error functions were obtained for the nonlinearized Langmuir model, confirming the same result predicted using correlation coefficient nonlinear analysis.

#### 2.3.2. Adsorbent Mass Effect

The effect of biosorbent mass on the adsorption of Pb(II) was explored using isotherm experiments. Figure 6 shows that the Pb(II) adsorption isotherms for doses of Raw-AC and AC-OH that varied from 5 to 20 g/L. The experimental data graphs were plotted alongside nonlinear Langmuir and Freundlich isotherm models.

The removal percentage of Pb(II) was found to increase by increasing the biosorbent dose and decrease with increasing of initial concentration of Pb(II). For example, at C_i_ Pb(II) = 500 mg/L, the removal percentages of Pb(II) by Raw-AC and AC-OH were found to be 52.59–89.47%, 76.04–99.83%, and 86.07–99.76% at a doses of 5, 15, and 20 g/L, respectively. This result can be explained by the increase in the number of adsorption sites with increasing biosorbent mass [31].

Experimental equilibrium adsorption data were accessed using the nonlinear forms of Langmuir, Freundlich, and Dubinin–Radushkevich isotherm models. Table 5 shows the parameters of the isotherm models and their respective correlation coefficients (R^2^).

Based on the correlation coefficient shown, for all Raw-AC dose values, the nonlinear Langmuir isotherm model showed a better fit to the adsorption data than the Freundlich and Dubinin–Radushkevich isotherm models. The maximum adsorption capacities (q_max_) were 53.7 mg/g, 53.5 mg/g, and 50.1 mg/g for 5 g/L, 15 g/L, and 20 g/L of Raw-AC, respectively. The decrease in the adsorption amount with increasing biosorbent concentration has been reported in various investigations, which can be attributed to the declining total biosorbent surface area caused by the aggregation of sorption sites [41].

At biosorbent concentrations of 5 g/L and 15 g/L, experimental adsorption isotherms of AC-OH were better fitted to the Langmuir and Dubinin–Radushkevich models, whereas at a higher biosorbent concentration (20 g/L), the biosorption data were better fitted using the Freundlich and Dubinin–Radushkevich models.

#### 2.3.3. Effect of Contact Time

Figure 7a,b show the uptake of Pb(II) versus contact time with Raw-AC and AC-OH, respectively. It was observed that for all initial Pb(II) concentrations ranging from 25 mg/L to 250 mg/L, between 80 and 85% of Pb(II) was removed in the first 10 min of contact time. Following this time, the Pb(II) removal rate became negligible, and equilibrium was achieved in 30 min for the biosorption of Pb(II) by both biosorbents. The rapid uptake of lead ions in the first stage, which is commonly observed for biosorption of metal ions, is due to physicochemical interactions between the metal solution and available functional groups on the biomass surface [42]. As can be seen from Figure 7a,b, the equilibrium time of the biosorption process was 30 min, and after this time, the insignificant removal rate of Pb(II) could be attributed to the saturation of the available sites on the Raw-AC and AC-OH biosorbents surfaces [43].

Table 6 shows the values of kinetic parameters of Pb(II) biosorption on Raw-AC and AC-OH biosorbents obtained from the nonlinear pseudo-first- (PFO) and pseudo-second-order (PSO) models (non-linear equations of PFO and PSO are shown in Appendix A).

Comparing the values of the determination coefficient (R^2^) and the q_e,calc_ obtained from both kinetic models, it can be observed that for all initial concentrations (Ci(Pb(II) = 25–250 mg/L), the nonlinear PSO model fit better than the PFO model. Furthermore, the PSO rate constants (k_2_) were found to decrease from 0.084–0.484 to 0.024–0.148 (g/mg min) for Raw-AC and from 0.488–3.175 to 0.022–0.032 (g/mg min) for AC-OH, for an increase in the concentration of Pb(II) from 25 to 250 mg/L. Biomass treated with NaOH showed a higher uptake of Pb(II) and biosorption rate k_2_ compared with the raw biomass. The enhancement of biosorption capacity of treated *Ardisia compressa* K. leaves was also observed in other biosorbents [32,44].

As shown in Table 6, the regression coefficients obtained from the nonlinear form of the PSO and PSO models were higher than 0.941; furthermore, the q_e,cal_ obtained by both kinetic models were similar to q_e,exp_. Therefore, the comparison of R^2^ and q_e,cal_ resulted in inadequate criteria for defining the best-fitting kinetic model. However, the values calculated by the six function errors for all initial concentrations of Pb(II) resulted in lower PSO than the PFO (Table 7), confirming that the PSO was the best model to fit the kinetic data. This result suggests that the removal of Pb(II) was controlled by chemisorption, which involved the strong surface complexation of metal ions with the functional groups on the surface of biosorbents [31].

#### 2.3.4. Temperature Effect on Sorption Capacity

The study of the temperature effect on the sorption of Pb(II) was evaluated using adsorption isotherm experiments by varying the temperature from 300.15 to 330.15 K. Figure 8 shows the adsorption isotherms for Raw-AC and AC-OH at T = 300.15, 315.15, and 330.15 K. It can be observed that the adsorption capacity of Pb(II) of Raw-AC and AC-OH was higher at room temperature compared with that at 315.15 and 330.15 K, confirming that the adsorption was an exothermic process. The decrease in adsorption capacity with temperature can be explained by the breaking of adsorption forces between the metal ions and active sites on the biosorbents caused by the excess of energy supplied to the system, which promoted the desorption of metal ions from surface biosorbent [45,46]. The exothermic adsorption of Pb(II) ions using various types of sorbents, such as *Azadirachta indica* [47], montmorillonite [45], rice husk [48], and activated tea waste [49], was previously reported.

The isotherm parameters were calculated using the nonlinear equations of Langmuir, Freundlich, and Dubinin–Radushkevich isotherm models.

As shown in Table 8, the adsorption of Pb(II) on AC-OH on the surface of different temperatures studied showed a best fitting to the Langmuir model, as indicated by the high correlation coefficient (R^2^ = 0.959–0.990) when the temperature increased from 300.15 K to 330.15 K, the Langmuir constant Q_m_ of AC-OH, which indicates the monolayer saturation at equilibrium, decreased from 171.0 to 45.3 mg/g. In the case of Raw-AC, the adsorption of Pb(II) at ambient temperature showed a good fit to the Langmuir model, whereas at T = 313.15 and 330.15 K, it was best fitted to the Freundlich model and the values of K_F_ and n parameters decreased with the temperature rise. At all temperatures studied, the values of n parameters were between 1 and 10, indicating that AC-OH and Raw-AC favorably adsorb Pb(II) ions.

Error functions for all temperatures studied were calculated. Table 9 shows that for AC-OH, the Langmuir isotherms exhibit lower values of error functions for all temperatures studied and is a better fit compared with the Freundlich isotherm.

### 2.4. Adsorption and Desorption Study

Figure 9a and Figure 9b show the adsorption and desorption cycles of Pb(II) on Raw-AC and AC-OH, respectively. It can be seen that after the three cycles, the percentage of Pb(II) removal on Raw-AC and AC-OH decreased from 80.82% to 45.70% and from 93.80% to 47.05%, respectively. Similarly, the Pb(II) desorption decreased from 67.75 to 45.70% and from 46.94% to 32.25% for Raw-AC and AC-OH, respectively. The decrease in the performance of adsorption and desorption of Raw-AC and AC-OH biosorbents was previously reported in other biosorbents, which was attributed to the possible remanent of metal ions on the surface of the biosorbent after the desorption step and the partial destruction of active functional groups by HCl after each regeneration cycle [22]. Despite the reduction in the percentage of Pb(II) removal after three cycles, both biosorbents have still shown efficient removal and desorption of Pb(II) after three times of recycling. As such, Raw-AC and AC-OH can be considered suitable recyclable biosorbents.

### 2.5. Comparison of Natural and Alkali Ardisia Compressa K. with Other Biosorbents

In this study, the maximum Pb(II) biosorption capacity, according to Langmuir’s model, was 96.4 and 170.9 mg/g using Raw-AC and AC-OH, respectively. As shown in Table 10, the results obtained in this work are within the usual values, and in some cases, they are between the highest maximum biosorption capacities for other similar biosorbents reported in the scientific literature, indicating the potential for its application in the removal of lead, and perhaps other heavy metals too, from contaminated effluents.

### 2.6. Proposed Mechanism of Interaction of Pb^2+^ with Raw-AC and AC-OH

The mechanism of interaction of Pb^2+^ with Raw-AC and AC-OH could be represented as shown in Figure 10 below:

According to the FTIR results, both biomasses (Raw-AC and AC-OH) present, OH-,-COOH, and NH_2_ could be complexed with Pb^2+^ species, or some ion exchange could occur. Adjusting the pH to 4–6, the COOH groups of carboxylic acid release H^+^ into the solution. This pH adjustment helps the complexation or electrostatic attraction of Pb^2+^ with both Raw-AC and AC-OH biomasses.

### 2.7. Data-Driven Optimization

An optimization study was conducted for the two materials (Raw-AC and AC-OH) and its objective was to find which type of material, along with its multivariate combination of initial concentration of Pb, mass, temperature, pH of the solution, and contact time, could provide the maximum q_e_. However, optimization using the experimentation data required a couple of steps which are mentioned herewith:In the first step: the original data was arranged in the form of a database (see Table 11), where all the tuning variables and the performance indicators were placed together. The database contained a total of 178 experimental points. Bringing the data in such a format would help in the development of an empirical model [61]. It was necessary to develop an empirical model because the optimization algorithms required a direct correlation between the tuning variables and the performance indicator. Thus, it was also necessary to understand the data through the heatmap of the Pearson coefficient of correlation (see Table 12) [62]. It was noted that the initial concentration of Pb was the variable with the highest level of sensitivity for q_e_, and it was also noted that material 2 had a relatively higher tendency for higher values of q_e_. The temperature and the pH of the solution were the least sensitive.

2.In the second step, an empirical model was developed for the selected database. Such a model was developed through the application of artificial neural networks for regression models. The MATLAB environment using the nftool box was implemented. All the input variables were placed in the input layer, and the performance indicator was placed in the performance indicator. A backpropagation method called Bayesian regularization backpropagation was implemented, which was suitable for the noisy database as the origin was experimentation [63]. The training, testing, and validation percentages were set to be 70, 15, and 15, respectively. The number of hidden neurons was iterated and a total of eight neurons came out to be optimal for the minimum mean square error. The regression fit and the error histogram are displayed in Figure 11. During the training, testing, and total phase, the values of R came out to be 0.99276, 0.977, and 0.9905, respectively, which indicates a high goodness of fit. The error histogram also fulfills the normality assumption of errors.

3.Once the empirical model was developed, in the last step, an optimization study could be conducted. The optimization study was conducted to maximize qe for the given range of each of the tuning variables. The genetic algorithm was implemented in the MATLAB environment through its Optimization toolbox. All the generic configuration of the algorithm was taken to be default [64]. The optimization was conducted two times, one for each material, and the results are shown in Table 13. It is noted that the combination of tuning variables for each type of material was quite similar. For example, for Raw-AC, the optimal combination of temperature, pH of the solution, and contact time came out to be the same, which was 298.15 K, 6, and 1440 min, respectively. The only difference was noted in the cases of initial concentration of Pb and mass, which were 854.16 mg/L and 0.1 g for Raw-AC and 1012.98 mg/L and 0.05 g for AC-OH, respectively. It is quite interesting that even though a couple of variables were the same, the value of optimal q_e_ was quite different. For the combination of Raw-AC and AC-OH, the optimal q_e_ values came out to be 62.287 mg/g and 147.475 mg/g, respectively. It is remarkable how doing such a treatment can improve the results by 57.76%. It is also to be stressed here that the optimal point evaluated through the computational method was placed back into the experimental conditions and conformity between the optimal point through the computational method and the experimentation was met.

## 3. Materials and Methods

### 3.1. Pb standard Solutions

Stock solutions (1000 mg/L) of Pb(II) were prepared by dissolving 0.337 g of PbCl_2_ (99%, Sigma Aldrich, St. Louis, MO, USA) in deionized water. Solutions with 25 to 700 mg/L concentrations were prepared by diluting the stock solutions using deionized water. Pb(II) calibration curves were obtained by diluting 1000 mg/L standard solutions of Pb in 2% *v*/*v* HNO_3_ (Fluka Analytical, St. Louis, MO, USA).

### 3.2. Biosorbents

Leaves of *A. compressa* K. were manually collected from the experimental garden at Universidad Tecnológica de Xicotepec de Juárez, located in Xicotepec de Juárez, Puebla (−97.96 20° 16′ 33′′ N, 97° 57′ 36′′ W). Fresh leaves of *A. compressa* K. were washed, disinfected, and air-dried. The dry material was ground in a blender until a homogeneous fine powder (particle size between 0.42 and 0.80 mm) was obtained. The powder was preserved until use at room temperature in hermetically sealed plastic bags in the absence of light and humidity.

Alkali treatment of raw *A. compressa* K. (Raw-AC) was done as follows: 10 g of pulverized Raw-AC was added to 250 mL of 1.0 M NaOH in an Erlenmeyer flask (500 mL) and shaken at 140 rpm for 2 h. Then, the sample was filtered under a vacuum and rinsed several times using deionized water until the wash solution pH reached approximately 6.5. The treated biomass labeled AC-OH was air-dried and stored in glass bottles. Figure 12 illustrates the preparation procedure of the biosorbents.

### 3.3. Physicochemical Analysis

Association of Analytical Communities (AOAC (2007)) methods were used for the chemical characterization of *A. compressa* leaves [65]: ash, moisture, raw fiber, and ethereal extract were found using the Soxhlet method; protein was found using the Kjeldahl method and nitrogen-free extract (NFE) was calculated using the percentage differences. The total soluble solids (TSSs) were identified using a digital refractometer (PR-101ATAGO PALETTE) according to the method previously described [65]. All determinations were done in triplicate.

### 3.4. Characterization Techniques

The pH_PZC_ was obtained as previously described [32]. Samples of Raw-AC and AC-OH, before and after contact with metal ions, were characterized using a Fourier transform infrared spectrometer (Nicolet Nexus 670) to identify the functional groups on the biomass surface. Surface characteristics of Raw-AC and AC-OH biosorbents before and after the uptake of Pb(II) were studied using scanning electron microscopy (JEOL JSM-IT300). An energy-dispersive X-ray spectrometer was used to determine the approximate chemical composition on the surface of Raw-AC and AC-OH biosorbents.

### 3.5. Biosorption Study

In this work, the biosorption process was carried out using a batch technique. For the kinetic study, 0.1 g of Raw-AC and AC-OH was mixed with 10 mL of Pb(II) solutions at initial concentrations from 25 mg/L to 250 mg/L. The samples were shaken at 140 rpm in an orbital shaker (CRP-0228, Scientific) for times ranging from 15 min to 1440 min. After each contact time, the samples were centrifuged (CRM Globe, Certificient, Chicago, IL, USA) for 5 min, and the Pb(II) contents in the aqueous solutions were determined using flame atomic absorption spectrometry (iCe Serie TermoScientific).

After each contact time, the amount of Pb(II) adsorbed on the surface of Raw-AC and AC-OH was calculated using Equation (1):(1)$$q_{t}=\frac{C_{i}-C_{t}}{m}\cdot V\;$$
where *C_i_* and *C_t_* are, respectively, the initial and final concentrations of Pb(II) (mg/L); *V* (L) is the volume of solution; and m (g) is the mass of biosorbents.

In this study, the effects of the solution pH, dose of the biosorbents, and temperature were carried out by varying the initial lead concentration from 25 to 1000 mg/L. For the kinetic study, the temperature and solution pH were fixed, whereas the initial lead concentration was varied from 25 to 250 mg/L and the contact time was changed from 5 to 1440 min.

Table 14 gives the experimental conditions used to investigate the effects of contact times, solution pH, temperature, and dose of biosorbents on the removal of Pb(II).

All experiments were performed in duplicate and the results were expressed as averaged values. The Appendix A provide the kinetic and isotherm models, as well as the error functions analysis used in this study.

### 3.6. Kinetic Models

This study employed the nonlinear equations of pseudo-first- and pseudo-second-order models (Equation (2) and Equation (3), respectively) to investigate the kinetics of Pb(II) biosorption onto Raw-AC and AC-OH:(2)$$q_{t}=q_{e}\left(1-exp(-k_{1}t\right))$$
(3)$$q_{t}=\frac{k_{2}{q_{e}}^{2}t}{1+k_{2}q_{e}t}$$
where *q_e_* is the theoretical adsorption capacity (mg/g), and *k*_1_ (1/min) and *k*_2_ (g/mg min) are the pseudo-first- and pseudo-second-order rate constants, respectively.

### 3.7. Isotherm Models

Experimental data were examined using Langmuir, Freundlich, and Dubinin-Radushkevich (D-R) isotherm models.

Equation (4) describes the non-linear form of the Langmuir isotherm model:(4)$$q_{e}=\frac{q_{m}{K_{L}C}_{e}}{1+{K_{L}C}_{e}}\;$$
where *q_m_* (mg/g) is the maximum sorption capacity, and *K_L_* (L/mg) is the Langmuir equilibrium constant, which indicates the affinity of the sorbate for the solute [55].

The Langmuir model assumes that adsorption takes place for monolayer adsorption of the adsorbate from a liquid solution on a surface of biosorbent containing a finite number of identical sites and all adsorption sites have equal adsorption energies [55,66].

The Freundlich model is an empirical isotherm that can be used for non-ideal adsorption, which is mostly used to understand the adsorption of metal ions on a heterogeneous surface with multilayer adsorption. It can also define an exponential distribution of active sites and their energy [54]. The non-linear form of the Freundlich isotherm model is expressed as Equation (5):(5)$$q_{e}=K_{F}{C_{e}}^{1/n}\;$$
where *K_F_* (mg/g)(L/mg)^1/n^ and *n* are the Freundlich constants associated with the adsorption capacity and adsorption intensity of the adsorbent, respectively. *C_e_* (mg/L) is the adsorbate equilibrium concentration. When *n* = 1, the isotherm is linear and indicates that all sites on the adsorbent have equal affinity for the adsorbates. Values of *n* > 1 indicate the affinities decreased with increasing adsorption density [54]. The greater the value of n, the more favorable the adsorption and the more heterogeneous the surface of the particles will be [67].

The D-R isotherm is an empirical model that is generally employed to find whether the adsorption involved in the experimental data is physical or chemical [68]. The non-linear form of the D-R isotherm model is shown in the following Equation (6):(6)$$q_{e}=q_{m}e^{-\beta \epsilon^{2}}\;$$
where β (mol^2^/kJ^2^) is a constant related to adsorption energy, $\varepsilon =RTln(1+\frac{1}{C_{e}})$ is the adsorption potential (kJ mol^−1^), *R* (8.314 J mol^−1^ K^−1^) is the universal gas constant, *T* (K) is the absolute temperature, and $E=\frac{1}{\sqrt{2\beta }}$ is the adsorption energy (kJ/mol).

The numerical value from the free energy of the molecule (E) gives an indication of the nature of the interaction forces between lead ions and the active sites on the composite surface. When the value of the adsorption energy was less than 8 kJ/mol, it indicates that the adsorption occurred physically; if it was between 8 and 16 kJ/mol, it suggests that the adsorption process was carried out chemically, driven by the ion exchange mechanism; and chemisorption was observed when E values are between 20 and 40 kJ/mol [69].

### 3.8. Adsorption–Desorption Cycles

Experiments of adsorption and desorption cycles were conducted: 10 g/L of Raw-AC and AC-OH were mixed with 1000 mg/L of Pb(II) for 1440 min at room temperature and solution pH at 6. The samples were filtered, and the adsorption efficiency was obtained utilizing Equation (7):(7)$$Adsorption\;efficiency\;\%=\frac{C_{ads}}{C_{initial}}\times 100\;$$

After the adsorption process, the biosorbents were dried and weighed; then, the desorption experiments were conducted by mixing the biosorbents at ambient temperature for 2 h with 0.5 M HCl. The Pb(II) concentrations in each solution were measured, and the desorption efficiency was obtained utilizing Equation (8):(8)$$Desorption\;efficiency\;\%=\frac{C_{des}}{C_{ads}}\times 100\;$$
where *C_des_* is the concentration of Pb(II) desorbed in mg/g and *C_ads_* is the concentration of Pb(II) adsorbed in mg/g.

After desorption, the samples were rinsed several times with deionized water to remove the eluent agent from the biosorbents. The samples were then dried and weighed for further reuse. In this study, three adsorption and desorption cycles were repeated.

### 3.9. Framework of Empirical Model and Optimization Process

Owing to the complexity of the involvement of so many tuning variables, it is not easy to determine the optimal set conditions for this problem. For this purpose, an artificial intelligence method consisting of the artificial neural network was adapted for the empirical modeling, and later on, a genetic algorithm was applied for optimization purposes, which would eventually give out the best combination of all the multivariate tuning variables [70].

## 4. Conclusions

In conclusion, this study demonstrates that *Ardisia compressa* K. leaves, particularly those treated with a solution of 1M NaOH (AC-OH), exhibit a higher sorption capacity for Pb(II) from aqueous solutions compared with natural *Ardisia compressa* K. leaves (Raw-AC). The surface modification of the biosorbent after the alkali treatment was confirmed via pH_PZC_ and FTIR analysis, which revealed the presence of carboxyl, hydroxyl, and amino groups responsible for Pb(II) binding.

The removal of Pb(II) was found to be dependent on various factors, including solution pH, contact time, biosorbent dose, and water temperature. The optimum experimental conditions for maximum Pb(II) removal by both biosorbents were at pH 6 and T = 300.15 K.

The kinetic data analysis indicated that the pseudo-second-order kinetic model best described Pb(II) biosorption on both Raw-AC and AC-OH, which suggests that chemisorption controlled the rate of reaction. The Langmuir isotherm model described the adsorption isotherms for both biosorbents at different solution pH values well, suggesting monolayer sorption on a homogeneous surface took place. Furthermore, the study introduced the application of artificial neural networks and genetic algorithms for regression modeling and optimization purposes.

The model showed high reliability, and the optimization study revealed a significant improvement of around 57.76% in the indicator “q_e_” for AC-OH. The optimal conditions for AC-OH were determined as follows: initial concentration of Pb: 1012.98 mg/L, mass: 0.05 g, temperature: 298.15 K, pH of the solution: 6, and contact time: 1440 min, resulting in an optimal q_e_ of 147.475 mg/g. In summary, the results indicate that *Ardisia compressa* K. leaves can serve as a promising, cost-effective, and efficient biosorbent for removing Pb(II) ions from aqueous solutions. The study also highlights the potential of artificial neural networks and genetic algorithms for optimizing biosorption processes. These findings contribute to the development of environmentally friendly and sustainable methods for heavy metal removal from water sources.

A significant finding of this study pertains to the coefficient of determination, which indicated that the Dubinin–Radushkevich isotherm model exhibited lower values compared with the Langmuir and Freundlich, suggesting a need for further investigation into the applicability and parameter-based assessment of the Dubinin–Radushkevich model to elucidate sorption mechanisms.

The insights from this research hold valuable implications for engineering education, as it showcases how modifying biosorbents, optimizing process conditions, employing kinetic and isotherm models, and utilizing advanced techniques like artificial neural networks and genetic algorithms can be integrated into engineering curricula to foster a comprehensive understanding of designing effective and eco-friendly solutions for heavy metal removal in water treatment systems. The social insights derived from this research underscore the potential impact of innovative engineering approaches on addressing environmental challenges, demonstrating the significance of interdisciplinary collaboration between engineering and environmental sciences to develop sustainable solutions for water pollution, thereby imparting the importance of considering broader societal and ecological implications of technological advancements in engineering education.

## Figures and Tables

**Figure 1 molecules-28-06387-f001:**
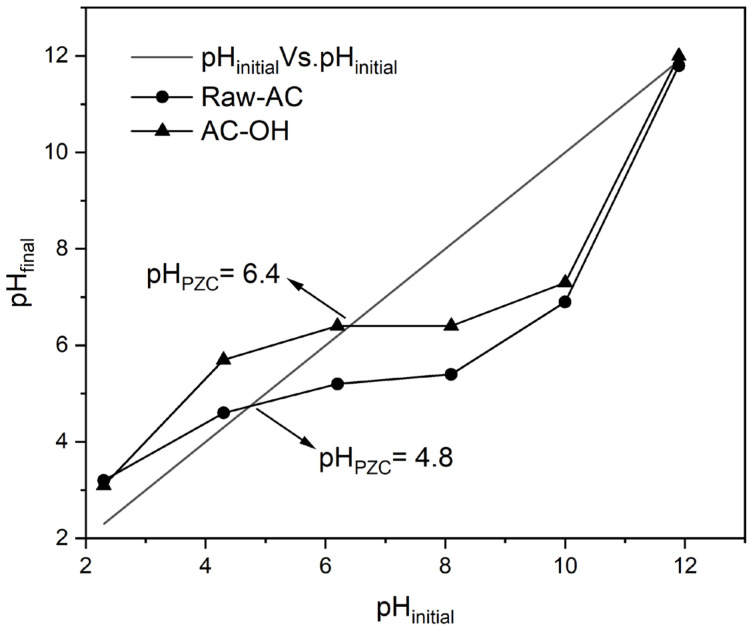
pH_initial_ vs. pH_final_ of Raw-AC and AC-OH biosorbents.

**Figure 2 molecules-28-06387-f002:**
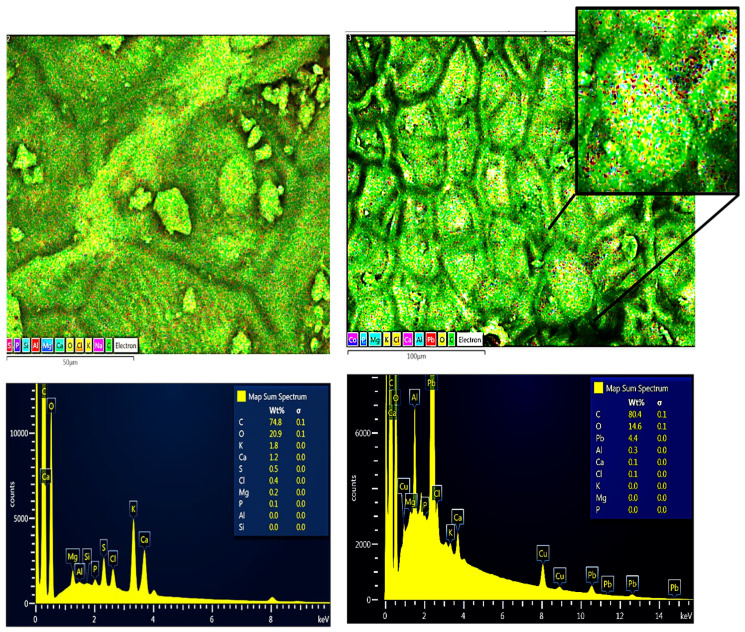
SEM images and EDS analysis of RAW-AC (**left**) and RAW-AC--Pb (**right**); in the zoomed-in image, the red dots represent the Pb(II) on the surface of the sample.

**Figure 3 molecules-28-06387-f003:**
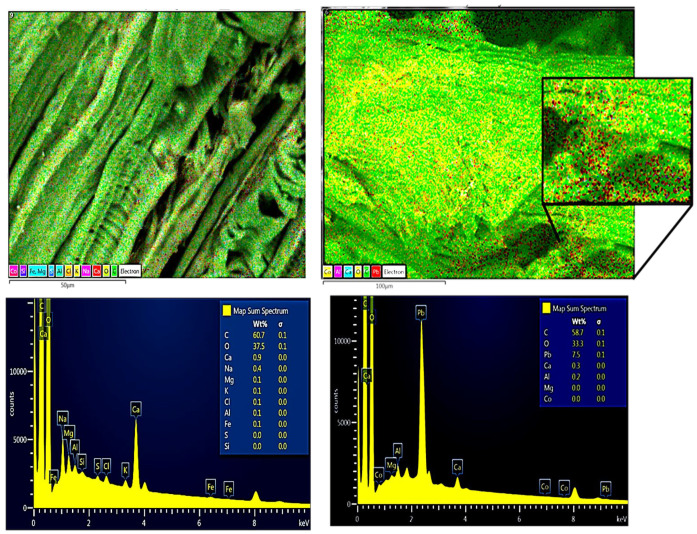
SEM images and EDS analysis of AC-OH (**left**) and AC-OH-Pb (**right**); in the zoomed-in image, the red dots represent the Pb(II) on the surface of the sample.

**Figure 4 molecules-28-06387-f004:**
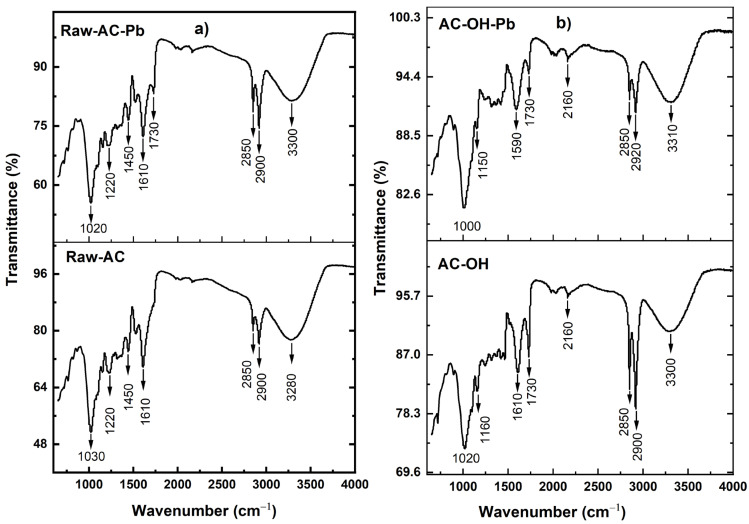
FTIR spectra of (**a**) Raw-AC and Raw-AC-Pb and (**b**) AC-OH and AC-OH-Pb.

**Figure 5 molecules-28-06387-f005:**
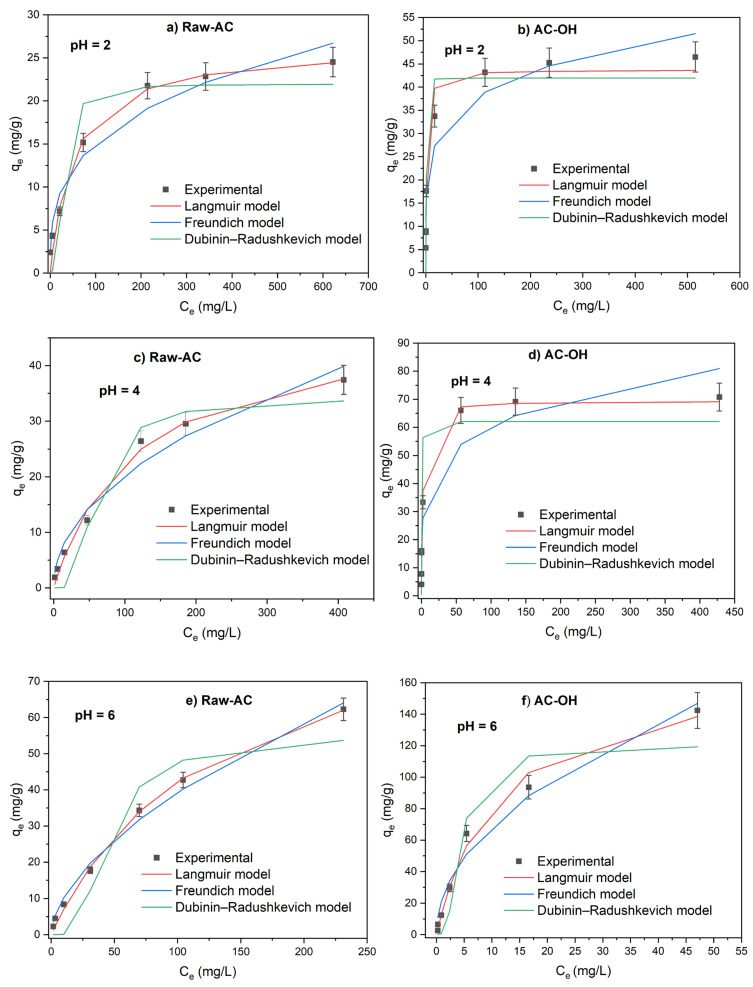
Adsorption isotherms of Pb(II) on Raw-AC and AC-OH at pH = 2 (**a**,**b**), pH = 4 (**c**,**d**), and pH = 6 (**e**,**f**).

**Figure 6 molecules-28-06387-f006:**
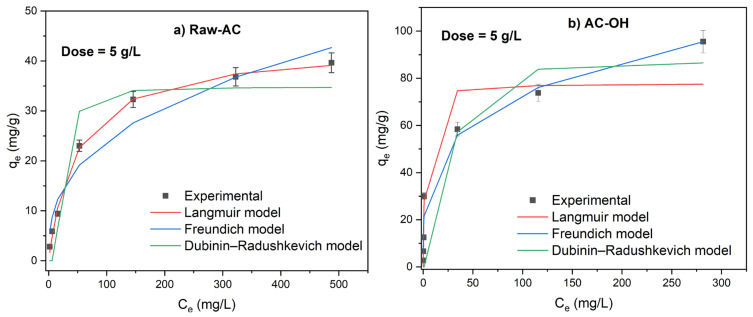
Adsorption isotherms of Pb(II) on Raw-AC and AC-OH at doses of 5 g/L (**a**,**b**), 15 g/L (**c**,**d**), and 20 g/L (**e**,**f**).

**Figure 7 molecules-28-06387-f007:**
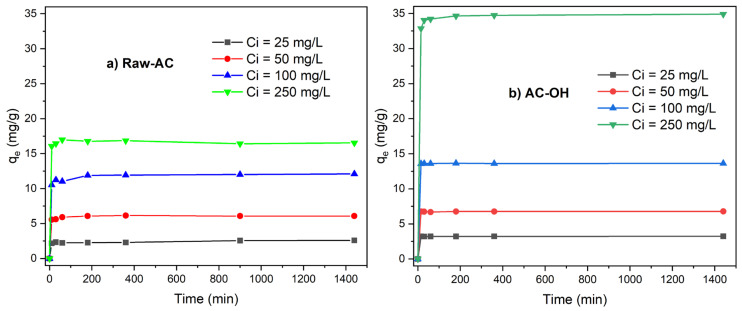
q_e_ (mg/g) vs. contact time t (min) of (**a**) Raw-AC and (**b**) AC-OH.

**Figure 8 molecules-28-06387-f008:**
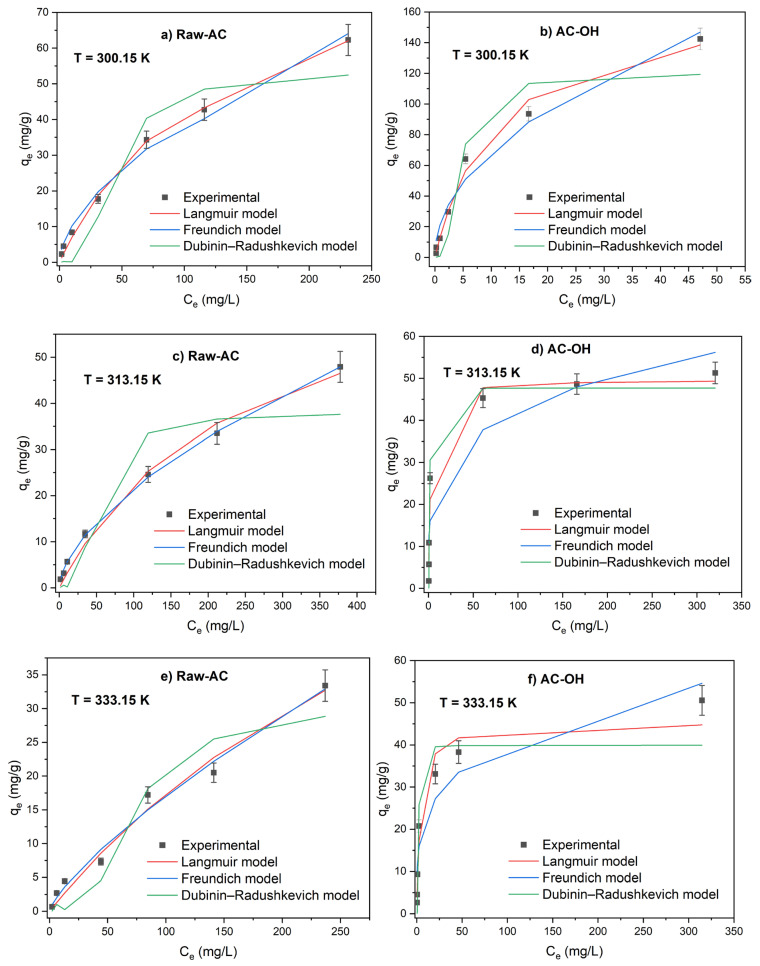
Adsorption isotherms of Pb(II) on Raw-AC and AC-OH at temperatures 300.15 (**a**,**b**), 313.15 (**c**,**d**), and 330.15 K (**e**,**f**).

**Figure 9 molecules-28-06387-f009:**
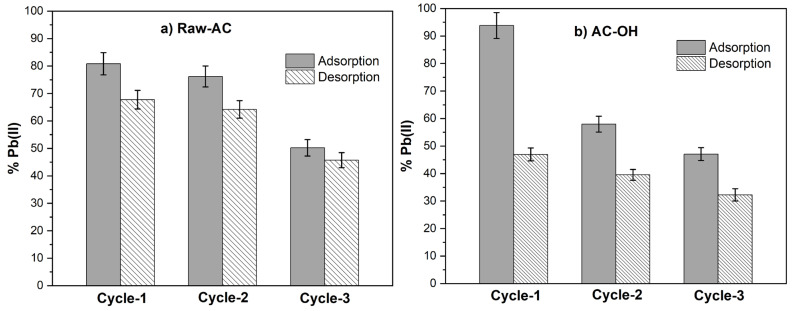
Adsorption and desorption cycles of Pb(II) with (**a**) Raw-AC and (**b**) AC-OH biosorbents.

**Figure 10 molecules-28-06387-f010:**
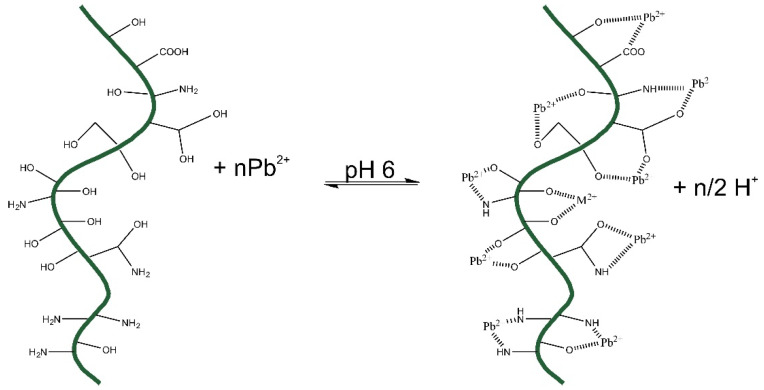
Interaction mechanism of the uptake of Pb^2+^ with raw-AC and AC-OH.

**Figure 11 molecules-28-06387-f011:**
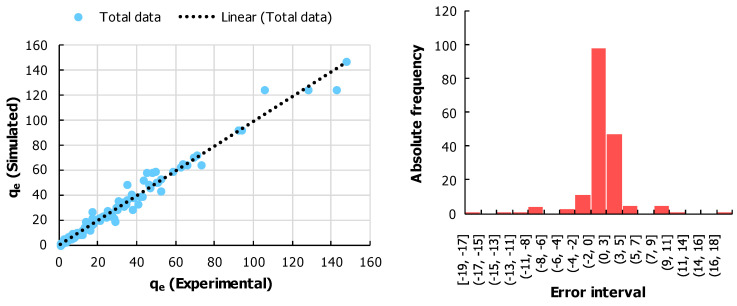
(**Left**) Results of empirical modeling through a comparison between experimental and simulated q_e_. (**Right**) Histogram of the errors between the experimental and simulated q_e_ values.

**Figure 12 molecules-28-06387-f012:**
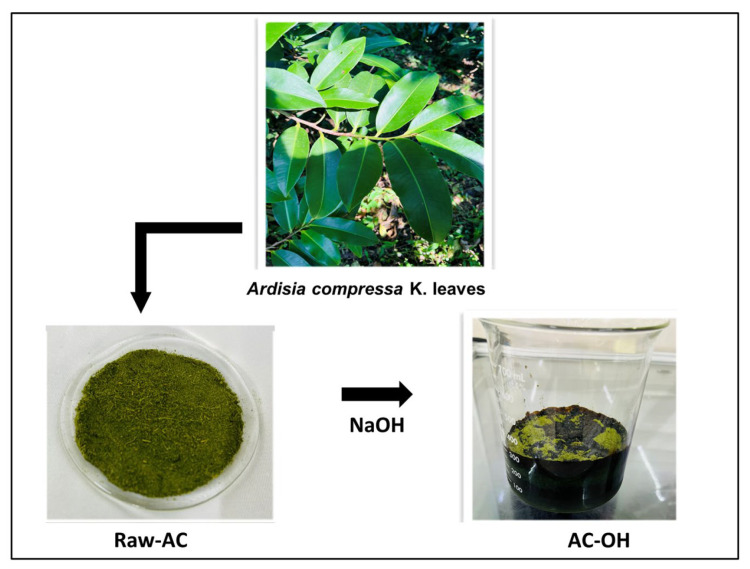
Schematic representation of the preparation of the biosorbents.

**Table 1 molecules-28-06387-t001:** Physicochemical parameters of *Ardisia compressa* K. leaves.

Parameter	Percent Composition (% *w*/*w*)
Total soluble solids (%)	86.49 ± 0.12
Moisture (%)	13.51 ± 0.12
Ash (%)	6.74 ± 0.01
Ethereal extract (%) Raw fiber (%) Protein (%)	3.60 ± 0.29 16.51 ± 0.34 5.83 ± 0.12
Nitrogen-free extract (%)	53.80 ± 0.36

All values are reported in % on a dry weight basis.

**Table 2 molecules-28-06387-t002:** Comparison of pH_PZC_ values of untreated and alkaline treated biosorbents.

Biosorbent	pH_PZC_ (Raw Biosorbent)	pH_PZC_ (Alkali-Treated Biosorbent)	Ref.
*Prunus armeniaca* L. *shells*	4.9	5.7	[31]
*Leucaena leucephala leaves*	6.7	7.2	[32]
*Cupressus sempervirens* *Carob shells* *Nostoc commune*	6.1 5.4 1.3	6.7 6.52 2.5	[33] [34] [28]
*Ardisia compressa* K.	4.8	6.4	This study

**Table 3 molecules-28-06387-t003:** Nonlinear Langmuir, Freundlich, and Dubinin–Radushkevich isotherm parameters for Pb(II) biosorption on Raw-AC and AC-OH biomass at solution pH = 2, 4, and 6.

pH	Model	Parameters	Raw-AC	AC-OH
2	Langmuir	Q_max_ (mg/g)	26.4	43.7
		K_L_·10^−2^ (L/mg)	2.001	59.431
		R^2^	0.981	0.971
	Freundlich	K_F_ (mg/g)(L/mg)^1/n^	3.580	16.226
		n	3.200	5.404
		R^2^	0.960	0.924
	Dubinin–Radushkevich (D-R)	Q_max_ (mg/g)	21.948	41.965
		β (mol^2^/kJ^2^)	93.593	0.242
		E (kJ/mol)	0.073	1.437
		R^2^	0.897	0.917
4	Langmuir	Q_max_ (mg/g)	48.1	69.41
		K_L_·10^−2^ (L/mg)	0.882	56.634
		R^2^	0.992	0.984
	Freundlich	K_F_ (mg/g)(L/mg)^1/n^	2.229	23.975
		n	2.084	4.979
		R^2^	0.970	0.913
	Dubinin–Radushkevich (D-R)	Q_max_ (mg/g)	34.183	62.050
		β (mol^2^/kJ^2^)	413.006	0.100
		E (kJ/mol)	0.035	2.236
		R^2^	0.932	0.867
6	Langmuir	Q_max_ (mg/g)	96.4	171.0
		K_L_·10^−2^ (L/mg)	0.781	9.076
		R^2^	0.997	0.990
	Freundlich	K_F_ (mg/g)(L/mg)^1/n^	2.654	22.311
		n	1.710	2.043
		R^2^	0.992	0.976
	Dubinin–Radushkevich (D-R)	Q_max_ (mg/g)	55.216	120.205
		β (mol^2^/kJ^2^)	239.383	2.729
		E (kJ/mol)	0.046	0.428
		R^2^	0.910	0.913

**Table 4 molecules-28-06387-t004:** Error function values of the Langmuir, Freundlich, and Dubinin–Radushkevich isotherm models at solution pH = 2, 4, and 6.

Biosorbent	pH	Model	Error Functions					
			APE	SSE	∆q(%)	χ^2^	EABS	RMSE
Raw-AC	2	Langmuir	9.934	4.703	19.201	1.790	5.898	0.970
		Freundlich	16.261	21.035	19.881	1.581	10.604	2.051
		D-R	36.483	51.896	57.739	353.992	15.899	3.222
	4	Langmuir	12.051	8.054	17.513	1.223	5.875	1.269
		Freundlich	19.734	36.164	23.840	2.180	13.922	2.689
		D-R	47.706	81.307	70.448	509.599	21.196	4.033
	6	Langmuir	16.786	9.119	30.325	3.133	6.721	1.510
		Freundlich	17.545	25.427	29.171	1.516	12.695	2.255
		D-R	53.582	272.070	72.248	903.793	41.273	7.377
AC-OH	2	Langmuir	6.402	54.695	8.453	3.009	14.617	3.307
		Freundlich	17.359	142.978	24.869	7.253	28.274	5.347
		D-R	30.443	156.436	47.782	911.672	30.073	5.593
	4	Langmuir	24.359	87.296	30.821	10.184	19.823	4.178
		Freundlich	68.640	469.134	43.902	16.675	52.871	9.686
		D-R	38.773	719.036	53.671	51.392	54.909	11.992
	6	Langmuir	16.701	169.569	21.784	4.109	25.363	5.824
		Freundlich	81.765	402.405	131.184	16.084	48.817	8.971
		D-R	55.164	1427.996	71.002	427.180	88.157	16.900

**Table 5 molecules-28-06387-t005:** Nonlinear Langmuir, Freundlich, and Dubinin–Radushkevich isotherm parameters for Pb(II) biosorption on Raw-AC and AC-OH biomass at dose = 5 g/L, 15 g/L, and 20 g/L.

Dose (g/L)	Model	Parameters	Raw-AC	AC-OH
5	Langmuir	Q_max_ (mg/g)	53.7	73.7
		K_L_·10^−2^ (L/mg)	2.105	0.773
		R^2^	0.996	0.957
	Freundlich	K_F_ (mg/g)(L/mg)^1/n^	4.905	23.916
		n	2.494	4.414
		R^2^	0.953	0.953
	Dubinin–Radushkevich (D-R)	Q_max_ (mg/g)	34.771	87.074
		β (mol^2^/kJ^2^)	68.942	81.735
		E (kJ/mol)	0.085	0.078
		R^2^	0.908	0.836
15	Langmuir	Q_max_ (mg/g)	53.5	53.3
		K_L_·10^−2^ (L/mg)	0.613	128.915
		R^2^	0.996	0.966
	Freundlich	K_F_ (mg/g)(L/mg)^1/n^	0.952	24.446
		n	1.539	2.829
		R^2^	0.986	0.964
	Dubinin–Radushkevich (D-R)	Q_max_ (mg/g)	29.303	47.476
		β (mol^2^/kJ^2^)	310.592	0.136
		E (kJ/mol)	0.040	1.916
		R^2^	0.914	0.905
20	Langmuir	Q_max_ (mg/g)	50.1	90.6
		K_L_·10^−2^ (L/mg)	0.801	30.554
		R^2^	0.998	0.993
	Freundlich	K_F_ (mg/g)(L/mg)^1/n^	0.941	20.674
		n	1.472	1.395
		R^2^	0.989	0.997
	Dubinin–Radushkevich (D-R)	Q_max_ (mg/g)	25.706	42.859
		β (mol^2^/kJ^2^)	157.144	0.212
		E (kJ/mol)	0.056	1.537
		R^2^	0.955	0.951

**Table 6 molecules-28-06387-t006:** Parameter values calculated using the nonlinear forms of the pseudo-first-order and pseudo-second-order kinetic models for Pb(II) biosorption on Raw-AC and AC-OH biomass at C_i_ = 25 mg/L, 50 mg/L, 100 mg/L, and 250 mg/L.

Biosorbent	Ci (mg/L)	Pseudo-First-Order			Pseudo-Second-Order		
		q_e,cal_ (mg/g)	k_1_ (1/min)	R^2^	q_e,cal_ (mg/g)	k_2_ (g/mg min)	R^2^
Raw-AC	25	2.4	0.341	0.976	2.4	0.970	0.994
	50	6.0	0.118	0.985	6.2	0.038	0.987
	100	11.7	0.228	0.941	11.9	0.058	0.973
	250	16.7	0.332	0.990	16.7	0.148	0.991
AC-OH	25	3.2	0.409	0.999	3.2	3.175	0.999
	50	6.7	0.298	0.999	6.8	0.728	0.999
	100	13.6	0.232	0.999	13.7	0.153	0.999
	250	34.5	0.200	0.997	34.8	0.032	1.000

**Table 7 molecules-28-06387-t007:** Error functions of the PFO and PSO kinetic models.

Biosorbent	Ci (mg/L)	Model	Error Functions					
			ARE	SSE (10^−2^)	∆q(%) (10^−2^)	χ^2^ (10^−2^)	EABS	RMSE
Raw-AC	25	PFO	1.486	1.531	2.099	0.635	0.251	0.055
		PSO	1.285	1.065	1.728	0.444	0.218	0.046
	50	PFO	1.758	9.475	2.218	1.633	0.698	0.138
		PSO	1.725	8.188	2.166	1.510	0.663	0.128
	100 250	PFO	2.790	99.482	3.577	8.505	2.252	0.446
		PSO	1.712	44.854	2.446	3.874	1.360	0.300
		PFO	1.051	28.115	1.297	1.688	1.226	0.237
		PSO	0.933	26.359	1.253	1.583	1.089	0.230
AC-OH	25	PFO	0.310	0. 093	0.424	0.029	0.060	0.015
		PSO	0.298	0. 064	0.352	0.019	0.057	0.013
	50	PFO	0.318	0.552	0.495	0.082	0.128	0.037
		PSO	0.295	0.459	0.452	0. 068	0.119	0.034
	100 250	PFO	0.136	0.446	0.220	0.033	0.111	0.033
		PSO	0.341	1.560	0.414	0.116	0.276	0.062
		PFO	0.711	46.737	0.889	1.355	1.467	0.342
		PSO	0.263	6.943	0.345	0.204	0.539	0.132

**Table 8 molecules-28-06387-t008:** Langmuir, Freundlich, and Dubinin–Radushkevich (D-R) isotherm parameters using nonlinear regression analysis for Pb(II) biosorption on Raw-AC and AC-OH biomass at different temperatures.

Temperature (K)	Model	Parameters	Raw-AC	AC-OH
300.15	Langmuir	Q_max_ (mg/g)	96.4	171
		K_L_·10^−2^ (L/mg)	0.781	9.076
		R^2^	0.997	0.990
	Freundlich	K_F_ (mg/g)(L/mg)^1/n^	2.654	22.311
		n	1.71	2.043
		R^2^	0.992	0.976
	Dubinin–Radushkevich (D-R)	Q_max_ (mg/g)	53.840	120.205
		β (mol^2^/kJ^2^)	227.720	2.729
		E (kJ/mol)	0.047	0.428
		R^2^	0.899	0.913
313.15	Langmuir	Q_max_ (mg/g)	76.1	49.6
		K_L_·10^−2^ (L/mg)	0.417	42.48
		R^2^	0.988	0.971
	Freundlich	K_F_ (mg/g)(L/mg)^1/n^	1.375	14.096
		n	1.671	4.173
		R^2^	0.999	0.884
	Dubinin–Radushkevich (D-R)	Q_max_ (mg/g)	38.081	47.671
		β (mol^2^/kJ^2^)	264.046	0.324
		E (kJ/mol)	0.044	1.243
		R^2^	0.865	0.970
330.15	Langmuir	Q_max_ (mg/g)	92.4	45.3
		K_L_·10^−2^ (L/mg)	0.231	24.844
		R^2^	0.981	0.959
	Freundlich	K_F_ (mg/g)(L/mg)^1/n^	0.508	12.635
		n	1.311	3.329
		R^2^	0.985	0.904
	Dubinin–Radushkevich (D-R)	Q_max_ (mg/g)	30.879	39.923
		β (mol^2^/kJ^2^)	494.126	0.503
		E (kJ/mol)	0.032	0.997
		R^2^	0.904	0.887

**Table 9 molecules-28-06387-t009:** Error function values of the Langmuir, Freundlich, and Dubinin–Radushkevich equilibrium models at 300.15, 313.15, and 330.15 K.

Biosorbent	T (K)	Model	Error Functions					
			APE	SSE	∆q(%)	χ^2^	EABS	RMSE
Raw-AC	300.15	Langmuir	19.583	9.119	0.303	3.133	6.721	1.350
		Freundlich	20.027	20.996	0.291	1.405	11.534	2.049
		D-R	52.451	298.240	70.076	608.200	42.590	7.723
	313.15	Langmuir	26.940	20.969	0.397	7.404	11.441	2.048
		Freundlich	5.468	1.262	0.114	0.211	2.160	0.502
		D-R	52.210	244.613	67.720	210.523	35.096	6.994
	330.15	Langmuir	23.035	16.390	0.305	3.404	9.584	1.811
		Freundlich	18.858	12.375	0.242	1.310	7.947	1.573
		D-R	47.879	75.587	63.686	94.235	19.872	3.888
AC-OH	300.15	Langmuir	16.701	169.569	0.218	4.109	25.363	5.824
		Freundlich	81.765	402.405	1.312	16.084	48.817	8.971
		D-R	55.163	1427.996	71.002	427.158	88.157	16.900
	313.15	Langmuir	54.847	80.379	1.131	7.002	20.212	4.009
		Freundlich	102.467	317.822	2.177	19.917	40.802	7.973
		D-R	37.865	97.503	57.454	78.557	23.568	4.416
	330.15	Langmuir	18.713	84.480	0.244	3.088	20.839	4.110
		Freundlich	71.759	196.337	1.261	13.284	35.860	6.266
		D-R	43.927	231.236	59.709	10.934	35.336	6.801

**Table 10 molecules-28-06387-t010:** Comparison of Pb(II) adsorption capacity of different sorbents.

Biosorbent	Dose (g/L)	Ci (mg/L)	pH	Kinetic Model	Mechanism	q_e_,_max_ (mg·g^−1^)	Ref.
*Moringa oleifera tree leaves*	4	20–200	5	PSO	Chemisorption	34.6	[42]
*Nostoc commune*	0.5	127- 442	4.5–5.5	PSO	Chemisorption	384.6	[28]
*Cystoseira stricta*	1	0.5–1.0	3	-	-	65.0	[50]
*Leucaena leucocephala leaves*	10	10–500	6	PSO	Chemisorption	25.5	[51]
*Ceratophyllum demursum*	0.1	1.46–1.73	7	-	Physical, monolayer	44.8	[52]
*Punica geranatum leaves*	0.1	1.46–1.73	7	-	Physical, monolayer	31.78	[53]
*Mirabilis jalapa leaves*	0.1	50–125	4.5	-	Multimolecular layers	38.5	[54]
*Laurus nobilis leaves waste*	4	5–10	6	PFO	Chemisorption	96.1	[55]
*Cladophora fascicularis*	2	225.4–838.3	5	PSO	Chemisorption	227.7	[56]
*Nickel ferrite nanoparticle*	-	0.01–0.05	5	PSO	-	19.2	[57]
*MoS*_2_/*thiol-functionalized multiwalled* *carbon nanotube*	2	20–150	6	PSO	Multilayer	90.0	[58]
*Nano Zero-Valent Iron particles*	0.1	10–1000	6	PSO	Chemisorption	140.8	[59]
*Cu-Based Composite Track-Etched Membranes*	0.0033–0.0052	50	5	PSO	Chemisorption	0.44–0.56	[60]
*Raw-AC*	10	25–1000	6	PSO	Chemisorption	96.4	In this study
*AC-OH*	10	25–1000	6	PSO	Chemisorption	170.9	In this study

**Table 11 molecules-28-06387-t011:** Format of the database for the development of the empirical model. Note: material 1 refers to the combination of Raw-AC and material 2 refers to the combination of AC-OH.

Material	Initial Concentration of Pb (mg/L)	Mass (g)	Temperature (K)	pH of the Solution	Contact Time (min)	q_e_ (mg/g)
1	24.414	0.1	298.15	2	1440	2.41213
1	48.204	0.1	298.15	2	1440	4.33674
1	92.492	0.1	298.15	2	1440	7.1832
1	412.92	0.1	298.15	6	1440	34.3227
1	531.76	0.1	298.15	6	1440	41.574
1	854.16	0.1	298.15	6	1440	62.287
2	26.958	0.05	298.15	2	1440	5.36626
2	44.613	0.05	298.15	2	1440	8.84096
2	89.1	0.05	298.15	2	1440	17.61704
2	333.02	0.05	298.15	6	1440	63.7672
2	757.88	0.05	298.15	6	1440	127.848
2	1012.98	0.05	298.15	6	1440	147.475

**Table 12 molecules-28-06387-t012:** Understanding the correlation coefficient between the tuning and performance indicators through Pearson’s coefficient of correlation.

	Material	Initial Concentration of Pb	Mass	Temperature	pH of the Solution	Contact Time	q_e_
Material	1.00	−0.01	−0.24	0.01	−0.01	0.01	0.27
Initial concentration of Pb	−0.01	1.00	0.14	−0.02	−0.12	0.27	0.79
Mass	−0.24	0.14	1.00	−0.06	0.06	0.38	−0.18
Temperature	0.01	−0.02	−0.06	1.00	0.16	0.22	−0.03
pH of the solution	−0.01	−0.12	0.06	0.16	1.00	−0.23	−0.03
Contact time	0.01	0.27	0.38	0.22	−0.23	1.00	0.22
q_e_	0.27	0.79	−0.18	−0.03	−0.03	0.22	1.00

**Table 13 molecules-28-06387-t013:** Optimality of Raw-AC and AC-OH through conformity between the computational method and experimental procedure.

Biomass	Initial Concentration of Pb (mg/L)	Mass (g)	Temperature (K)	pH of the Solution	Contact Time (min)	Optimal q_e_ (mg/g)
Raw-AC	854.16	0.1	298.15	6	1440	62.287
AC-OH	1012.98	0.05	298.15	6	1440	147.475

**Table 14 molecules-28-06387-t014:** Values of contact times, solution pH, temperature, and adsorbent dose parameters used for the experimental process.

Experiments	Experimental Parameters				
	pH	Time (min)	Ci (mg/L)	Dose (g/L)	Temperature (K)
Solution pH	2–6	1440	25–1000	10	300.15
Dose	6	1440	25–1000	5–20	300.15
Contact times	6	5–1440	25–250	10	300.15
Temperature	6	1440	25–1000	10	300.15–330.15

## Data Availability

The data presented in this study are available on request from the corresponding author.

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
