# Peer review of "Biosorption of Pb(II) Using Natural and Treated Ardisia compressa K. Leaves: Simulation Framework Extended through the Application of Artificial Neural Network and Genetic Algorithm"

_molecules, 2023, doi:10.3390/molecules28176387_

Round 1
Reviewer 1 Report
This paper is quietly interesting and provides good chemistry experimental data and mathematical modelling. I'm sure this nice combination will be popular with readers.
Some notes have to be clarified by the authors in the revised version of the manuscript:
1. Data on collecting and preparation of plant materials is missing.
2. Introduction section should be elaborated. Please include current state-of-art and cite more recent papers on removing Pb(II) ions.
3. To confirm the presence of OH-groups in the AC-OH samples please provide XPS data with a discussion
4. The revised manuscript should add SEM images of studied samples.
5. Authors have to combine the spectra on Fig.3a and 3b (as well as 3c and 3d) for a comparative demonstration of samples with/without Pb
6. The main lack of this study – is the absence of discussions about the sorption mechanism. Authors must change the discussion and include all studied models in it, with as much as possible discussion about the adsorption mechanism. An explanation of which functional groups are responsible for Pb(II) ions removal is required. The authors recommended considering the adsorption mechanism in more detail and entering data on the adsorption isotherm according to the Dubinin-Radushkevich model. (please, use and cite next studies doi:10.3390/polym14194026, doi:10.3390/nano12193293). I’m sure it will help to discuss the sorption mechanism more clearly.
7. Comparative Table #10 is too general. Please specify in the revised version details of sorption experiments (amount of loaded sorbent, concentration of feed Pb(II) solution, pH). Please add data on the sorption kinetic (rate constant) and proposed sorption mechanism to this table. In this form revised Table will be more informative. I'm also suggesting adding not only biosorbents of Pb(II) ions in this table. Please add different types of sorbents (please cite next recent papers 10.3390/membranes13050495, 10.3390/w15020222, 10.1016/j.ceja.2022.100295, 10.1021/acsomega.9b01603, 10.1038/s41598-021-83363-1 et al.)
8. The conclusion section should be reduced and more specified.
9. List of references should be corrected according to the journal's style
Reviewer 2 Report
This paper presents an interesting study about the use of natural and chemically treated A. compressa K. leaves to remove lead from aqueous solutions. The authors carried out a characterization of their raw and chemically modified adsorbent, in order to study changes in physico-chemical properties. A large number of tests was carried out, as each parameter was studied using Langmuir and Freundlich isotherms. Numerous tests have enabled the development of a model for determining the conditions required to obtain the highest possible value for adsorption capacity. Some points related to the methodology could have improved the manuscript:
- The study is done only in batch with reconstituted water, at levels much higher than the lead pollution in real waters
- Other pollutants could have been added such as other metals (Cd, As, Ni, Cu, Hg, Cr…)
However, the topic is relevant for publication in Molecules, and the results obtained are interesting compare to the literature for pollutant removal from water by biosorbent. Consequently, I suggest to publish the article, after the following comments and corrections have been made.
General comments:
All along the manuscript: The adsorption of Pb on the surface versus onto the surface, please be consistent in the notation
The way to insert references (numbers or name of authors-be consistent)
It should be specified whether the tests were carried out with one, two or three replicates, and if more than one replicate was made, the error bars should be added to the graphs.
For adsorption capacities, values can be rounded (ie 53.7 mg/g instead of 53.675 mg/g).
Specific comments:
Abstract, L34: the authors write: “It was also found that AC-OH is more effective than Raw-AC in removing Pb(II) from aqueous solutions”, and L37, they write: The nonlinear Langmuir isotherm model was best fitted, and the maximum adsorption capacity of Raw-AC and AC-OH were 170 mg/g and 96 mg/g, respectively”. The maximum adsorption capacity is higher for Raw-AC, an inversion?
Introduction, L52: it may be useful to specify the real risks to which humans and animals are exposed (e.g. diseases).
Introduction, L55: it may be interesting to list the methods traditionally used to purify water and their major drawbacks. A recent review has been published about biosorption (https://doi.org/10.3390/toxics11050404)
Line 66: The authors explained the use of A. compressa K. for extraction of interesting components but do not explicitely explain why they select it for biosorption purpose.
Line 83: treated with NaOH
Line 91: if we compare
Line 91-94: the two sentences should be linked to avoid a sentence starting by “while”.
Line 100: “obtained” should be removed
Table 1: All numbers should be aligned. Values for nitrogen-free extract are not in the middle of the "percent composition" box in the table.
Line 105: initial should be in subscript In general in this paragraph, several errors like this one are present, and all along the manuscript (line 158, please put PZC in subscript)
Line 107: “indicates”
Table 2 can be in SI if necessary.
Zero-point charge: I do not really understand why increasing the pHPZC is better for biosorption purposes because natural waters are around 6.5-8.5 so the lower the pHPZC, the larger the range of pH of negatively charge biosorbent, isn’t it?
Line 116: Microscopy
As the composition given by the instrument depends directly on the part of the adsorbent analyzed, I recommend displaying here an average of several cartographies, carried out at different locations on the same sample.
Line 139: ester not esther
Figure 3: a b c d should be aligned and in d) there is a mistake for the band at 1150 not 2150 cm-1
Section 2.3.1 and 2.3.2: Langmuir and Freundlich models are applied but not presented, and few conclusions are reached (i.e. whether adsorption sites are homogeneous or heterogeneous).
Line 154: Why the authors did not study pH higher than 6 ? Some natural waters can be at higher pH. Also, a space is missing at pH =2.
Section 2.3.3: PSO and PFO models are applied but not presented, and few conclusions are reached (i.e. chimisorption or physisorption).
Line 156: predominant
Line 163: “sulfhydryl”? There is no indication in SEM or FTIR of such groups.
Line 165: R²
Line 169: a space is missing between 26.393 and 48.078.
Line 170: KL should be KL.
Line 171-172: 10-2
Line 173: I do not understand the meaning of the sentence, please rephrase
Line 175: The terms should be defined
Table 4: 56.6349.07 and 0.9920.9 are strange please verify
Section 2.3.2: I suggest detailing the variation in removal depending on initial lead concentration (giving removal rates, as when studying pH, for example).
Line 194: Ci
Line 203: good or better?
Line 204: qmax
Line 210-211: best or better?
Line 235: qe,calc : subscript? In general, check all along the manuscript
Line 236: better
Line 240: the raw biomass
Figures 4-5: add a b c d e f etc
Line 254: what do you mean by chemical reaction?
Line 260: it can be observed
Line 276: well fit
Table 9: values not Values
Line 296, 304: percentage
Figure 8: the authors should clarify how is calculated the desorption percentage: from the adsorbed Pb? It means some stay in the biosorbent even with HCl? How explain the difference between raw -AC and AC-OH for the 1st cycle of desorption?
Caption Table 11: Ac-OH
Line 345: “and the performance indicator 344 is placed in the performance indicator” please check
There are 2 Table 12
Line 369: remove “that”, and “the” same, optimal not Optimal
Line 371-2 should be rewritten
Line 378: stock solutions
Line 382: were collected
Line 403: before and after
Line 419: mass in g?
Part 2.6. I do not understand which set of data the authors use to build the model and with which data they validate it? Maybe a model validation step should be added? To do this, new trials are carried out experimentally and tested with the model, and experimental and theoretical results are compared, is that what you did?
Line 469: data were most represented by
Line 477: AC-OH
The conclusion should be revised to add the context of the study and the part “Framework of empirical model and optimization process”
It may be useful to review the experimental optimum conditions for lead adsorption on the two adsorbents (contact time, adsorbent concentration and initial lead concentration are missing).
References: Some DOI are missing
The supplementary material was not available for reviewing.
Some grammatical mistakes or typo should be corrected, please see the section comments to authors.
Round 2
Reviewer 1 Report
The authors responded satisfactorily to my queries, and the quality of the revised manuscript was significantly improved. But I'm still not satisfied with the explanation of the sorption mechanism. In my first report, I suggested the best way for the sorption mechanism explanation - the isotherm model of the Dubinin-Radushkevish. I suggested two recent papers where this model was explained. In the revised manuscript, the authors provided only a scheme and added some suggestions based on the FTIR data. But these explanations are not enough for a description of the sorption mechanism. The authors inserted suggested references in the revised manuscript but did not use useful information from these papers.
Thus, I'm still asking you to elaborate on this 2.6 sub-section and add missing information about the sorption mechanism based on the EDR value (physical sorption, chemisorption or ion-exchange mechanism).
Table 10 (line corresponded to the ref [60]) - Pb(II) concentration was 50 ppm - 50 mg/L. Please fix it.
Only after the corrections mentioned above queries this manuscript can be recommended for publication.
Author Response
Please see the ttachment.
Thank you
